# Evaluation of Asphalt Concrete Airport Pavement Conditions Based on the Airfield Pavement Condition Index (APCI) in Scope of Flight Safety

**Mariusz Wesołowski** *,† and **Paweł Iwanowski** †

Air Force Institute of Technology, ul. Ks. Boleslawa 6, 01-494 Warsaw, Poland; pawel.iwanowski@itwl.pl
* Correspondence: mariusz.wesolowski@itwl.pl; Tel.: +48-261-851-324
† These authors contributed equally to this work.

**Abstract:** Airoport infrastructure development requires care to maintain it in proper technical condition. Due to this, airport pavements should be constantly monitored, and, above all, correctly managed. High-level airport pavement management requires access to reliable information about their current technical condition as well as proper forecasting of this condition in the future. Obtaining good quality information about the technical condition of airport pavement should be based on a proven methodology, taking into account the introduced quality management system. The authors propose a method of technical pavement condition assessment based on the Airfield Pavement Condition Index (APCI), taking into account not only the results of the surface deterioration inventory, but also repair overviews, load bearing capacity, evenness and roughness of the surface, as well as the surface tensile bond strength. The method was developed during long-term work financed by the Ministry of Science and Higher Education. At the beginning of the article, the authors focus on reviewing the currently available methods of assessing the technical condition of the pavement. Then they briefly present the most popular surface assessment method based on the PCI indicator. Afterwards, a proprietary asphalt pavement assessment method based on the APCI indicator is proposed and an example of how to use the method is presented. Finally, they discuss the results and summarize the work done, and present further directions of work.

**Keywords:** airfield; asphalt pavement; management; maintenance; condition; deterioration; APCI

## 1. Introduction

Recent years have shown that dynamic development of air transport is inevitable. Passenger and freight transport is dynamically developing in the 21st century. Every year, more and more new air connections are created, and on already existing routes the frequency and occupancy of flights is increasing. The development of low-cost carriers and the development of the air network have a significant impact on this situation [1]. The Polish governmental institution related to aviation, which is the Civil Aviation Authority (ULC), predicts by 2030 the constant development of passenger air traffic in Poland at the level of 5% [2]. ULC analysis shows that, in 2030, air traffic will reach 60 million passengers a year in Poland alone. According to [3], the main factors in the dynamic development of air transport are the economic progress of the country, the increase in the wealth of the society, demographic development and the development of airport infrastructure. These factors seem to be stable nowadays, which means that the increase in air traffic seems not to be endangered either. The development of aviation is not only an increase in the number and frequency of connections. There are also more and more modern aircraft: both passenger and cargo, as well as military. Jet-powered aircraft deserve special attention here, as they require increasingly better-quality airport infrastructure.

The dynamic development of civil and military aviation determines the need for continuous development of airport infrastructure. A good example is the Central Communication Port (CPK) being built in Poland. The CPK is designed to take over a significant part of the air traffic from the Fryderyk Chopin Polish National Airport in Warsaw, which is on the verge of exhaustion. In addition, it is to provide the necessary infrastructure for emerging new air connections and will be a European transport hub. Airport infrastructure will be developed, and high levels of flight safety will be maintained. The safety of flight operations is crucial in terms of human health and life. A special case of airport pavements are those located at military airport facilities. Their constant operational readiness is crucial for national security.

Due to the safety of air operations in the ground maneuvering area, special attention should be paid to foreign objects that may occur on the surface, the so-called FOD (Foreign Object Debris). They are dangerous because of the possibility of their being sucked in by the operating engine, with either the aircraft chassis or plating being damaged. The aircraft engine is able to suck in elements large enough to destroy it. In Figure 1, a water whirlpool was created as a result of air sucked in by the F-16.

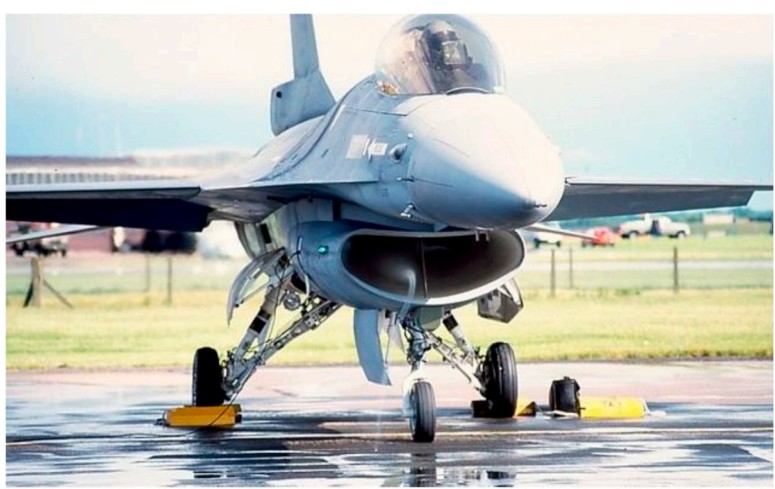

**Figure 1.** Water whirlpool being created as the result of air sucked in by the jet engine. Source: [4].

History proves many incidents and plane crashes are caused by the FOD. Some situations, such as the bollard suction by an aircraft engine [5], do not pose a serious threat if they are noticed in time. Larger objects can seriously damage engine blades and even get to the deeper parts. A view of engines with FOD sucked in is shown in Figure 2.

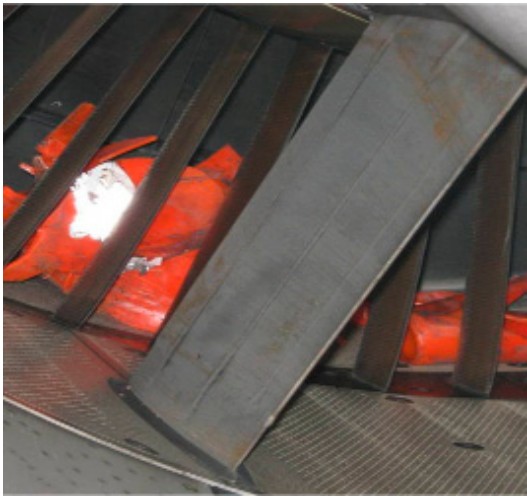 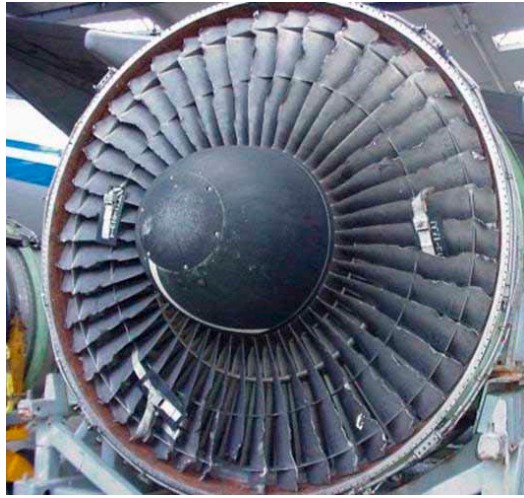

**Figure 2.** Elements sucked in by the engine. Source: [5,6].

Unfortunately, there have been situations in which foreign bodies posed lethal threat. One of the most tragic air catastrophes caused by sucking in a foreign object is the plane crash of the Air France Concorde F-BTSC aircraft, which, during take-off, drove over an object on the runway. As a result, the left wheel of the aircraft was damaged, and its fragments damaged the lower part of the wing. The aftermath was a tragic sequence of events that resulted in an engine fire and loss of thrust. Finally, the plane crashed near the airport, and 113 people were killed (including four on the ground) [7]. The view of the airplane taking off is shown in Figure 3.

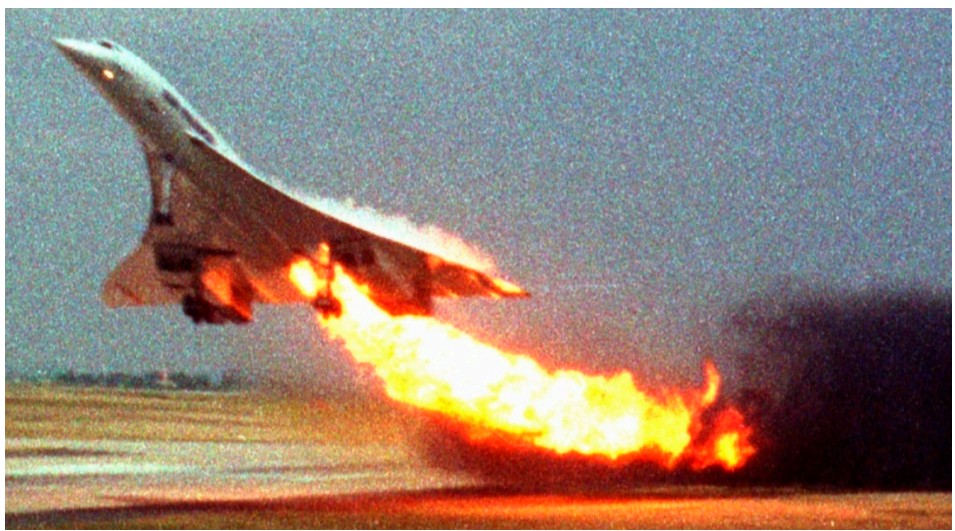

**Figure 3.** The burning engine with a bollard sucked in. Source: [7].

FOD may be, among other things, detached elements on the surface, which is why it is so important to ensure surface cleanliness and proper technical condition. Proper maintenance of airport pavements and ensuring their technical parameters at an appropriate level is one of the key factors that guarantees the safety of operations in the ground maneuvering field in the aspect of plane–airfield surface contact. The security of any processes can only be ensured based on a well-functioning quality management system. It is not different in the case of airport infrastructure management. The safety of operations in the ground maneuvering area depends on the airport pavement quality management system. Pavement management should not only be considered equivalent to actual pavement maintenance. Management should be approached comprehensively [8]. Effective management should involve the entire process, taking into account human resources, available equipment and materials as well as financial resources. The word effective should not only refer to the management of airfield surfaces here and now, but also the forecasting of future changes in the condition of the surface, enabling rational repair planning. Effective management also includes taking into account the life cycle of pavements planned for construction and those intended for future demolition or reconstruction.

A good-quality management system should be based on simple and legible procedures, clearly described and collected in the quality system's documentation. In the airport industry, there are documents created for the needs of managing both air traffic, flight safety and the technical condition of airport pavements. The basic document used all over the world is the Airport Services Manual issued by the International Civil Aviation Organization. The presented manual is divided into several parts, of which the second part describes how to deal with airport pavements. Many more documents have been created for public roads. In fact, every country, and even region, has its own road surface quality management documents. Examples of such systems can be found in [9–11]. In order to function properly, the quality system must be constantly monitored. In [9], the authors suggest conducting audits of pavement management systems. They also pay attention to controlling the proper functioning of the system in the context of collecting and transmitting information and reporting to higher organizational levels. Their review of the pavement management system (PMS) showed

that it contained three location subsystems, which made it difficult to combine data sets. In addition, attention was drawn to widely varying levels of data use by different districts and differences in maintenance operations. That is why it is so important to constantly supervise the implemented quality management system.

The book on airport management, road and parking surfaces [8] emphasized that surfaces should not only be maintained but also managed. Most of the costs incurred can be avoided by diagnosing the problem before the pavement is visibly deteriorated. This is related to the pavement's life cycle. Correct recognition of the current technical condition at a given moment of the pavement's life and forecasting this state in the future allows us to take actions much earlier than relying only on visual pavement assessment. Early response to changes in the technical condition of the pavement allows for a significant reduction in both financial and social costs. Figure 4 schematically shows the pavement life cycle in the context of its technical condition. In the first phase of pavement life, the degradation process is slow, and the costs of repairing any pavement damage are small and do not differ from one year to another. Over time, the dynamics of pavement degradation increase and reach such a level that year-by-year repair costs increase significantly. This is the time when major damage to the surface is already visible and its technical condition can be assessed as "FAIR". Repair costs can increase up to five times in a short surface life.

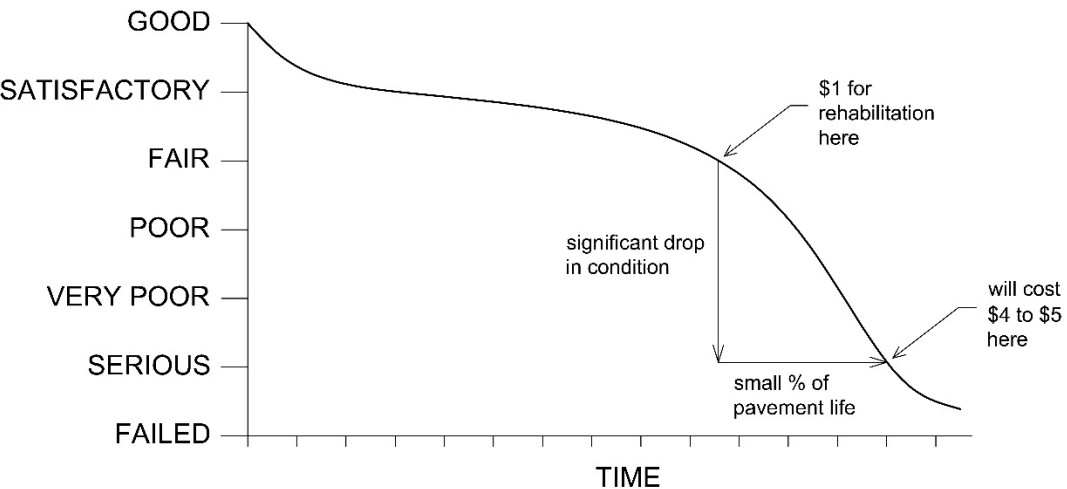

**Figure 4.** Schematic pavement life cycle. Source: [8].

Proper management of airport pavements, in particular forecasting their technical condition in the future, requires the use of appropriate tools based on reliable and current information on their current technical condition. Surface condition tests are conducted on the basis of test procedures. There are many methods around the world, almost all of which are derived from the Pavement Condition Index (PCI) procedure. This is a standard procedure developed and used by the US Army Corps of Engineers. At present, it is the most common research method among engineers, government institutions and organizations associated with airport pavements. One of the most popular procedures derived from the PCI method is the PAVER procedure used, among others, by [12] to assess the technical condition of 20 intersections in the Samsum region in Turkey. PAVER is an electronic system developed also by the US Army Corps of Engineers, enabling the collection and development of data obtained as a result of pavement review and their presentation in a graphic way on pavement diagrams. The Virginia Department of Transportation (VDOT) has developed the Distress Maintenance Rating (DMR), in which surface management is carried out on a "worst-first basis". According to this rule, pavement reconstruction schedules are created. The general approach to pavement assessment is based on the PCI method and the PAVER system, while VDOT has developed its own pavement condition indicators compatible with the conditions in Virginia [13,14]. A similar procedure was performed by the Italians [15]. They adopted the PCI damage catalog and adapted it to their needs and the

morphological conditions of the road surface. Two types of damage (manholes and tree roots) have been added and a new density/deduct curve has been added. In addition, the entire data processing cycle has been largely automated by implementing the application in a Visual Basic framework.

The PCI method is based only on visual assessment of the surface. This has the advantages of low cost and complexity in assessing the surface, but it also has some disadvantages. The most serious of these is the lack of comprehensive surface assessment. For this reason, many researchers have undertaken to expand the method with additional parameters. An example of such a solution may be the method proposed by [16], taking into account, apart from visual assessment of the surface, also the International Roughness Index (IRI) parameter. This parameter refers to the evenness of the surface, i.e., the surface properties describing the comfort of travel. Moreover, the authors presented the relationship between the IRI parameter and PCI, and also presented a model for estimating the PCI index based on measured IRI. Scientists from Kazakhstan showed their own method of assessing the condition of the pavement [17], who, besides the IRI index, also used elastic deflection measurement. In addition, they included macro-unevennessess, ruts and cracks. They proposed Pavement Condition Rating (PCR) as a parameter describing the surface conditions. On its basis, the surface is assessed in a functional context and it is the basis for making maintenance decisions. In India [18], a road pavement assessment method based on the Overall Pavement Condition Index (OPCI) was developed to manage the road network of the city of Noida. The OPCI model includes four indicators: Pavement Condition Distress Index, Pavement Condition Roughness Index, Pavement Condition Structural Capacity Index and Pavement Condition Skid Resistance Index. Each of the indicators is calculated individually, and then OPCI is calculated based on the above indicators.

Some of the proposed PCI surface assessment procedures automate the whole process somewhat. An example of such a procedure is in [19], where the authors developed a weighted approach to the PCI model. The method is based on a visual review of the surface, as is the case with the traditional PCI method, while the method of determining the final indicator is slightly different. Any damage enters the model with a suitable weight, and the entire process can be programmed. A similar approach was presented by [20] and [21]. To assess the degree of deterioration, they used data obtained as a result of damage and surface repair inventories. Both damage and surface repairs enter the model with characteristic weights. The weights are determined based on the harmfulness of the damage or repair.

In addition to information about the current technical condition of the pavement, an important role in managing airport infrastructure is played by the possibility of predicting the condition of the pavement in the future. The work in [22] presents attempts by Iranian scientists to develop an alternative method of determining the PCI indicator using optimization techniques based on artificial neural networks and genetic programming. This approach may, in the future, enable reliable extrapolation of the PCI indicator, and thus allow for rational planning of measures. Pavement management can also be based on an alternative method based on the Remaining Service Life (RSL) indicator describing the current and future condition of the airport pavement [23]. In Indonesia, on the other hand, the ANOVA statistical method was used to forecast RSL on the basis of PCI estimated for road surfaces in Sumatra [24]. A slightly different approach is used in the state of Indianapolis, where the PCI indicator is the basis for determining whether the surface has reached the minimum service life (MSL) level. This is the level which means enough time to take corrective actions on the given functional element of the airport [25].

Bearing in mind the importance of providing good quality data to the airport pavement management system, the authors propose an airport pavement assessment procedure based on the Airfield Pavement Condition Index (APCI). This indicator takes into account both pavement surface damage and repairs carried out in the past. In addition, information about the load-bearing capacity of the structure, anti-skid properties, longitudinal and transverse evenness of the pavement and the surface tensile bond strength are added to the model. The last of these parameters seems to be important, in particular, at airport facilities where jet-powered aircraft operate, especially due to the possibility of sucking in of foreign objects from the surface by the engine. This approach gives a

broad picture of the actual technical conditions of the airport pavement. This article is an extension of the information contained in [26] by the assessment of airport pavements made using asphalt concrete technology.

The main purpose of this work is to present a method for assessing the technical condition of asphalt airport pavements based on the APCI indicator, which is a reliable and practical tool supporting the operation of airport services in the field of managing airport pavements made of asphalt concrete.

After the introduction, containing an overview of existing solutions of airport pavement assessment, in the following sections, the authors will present a standard procedure for pavement assessment with the PCI method. Next, the procedure proposed by the authors based on the APCI indicator will be presented, along with a discussion of the basic research involved in obtaining input data. Finally, the authors will present a practical application, discuss the results and present the conclusions of this work.

## 2. Materials and Methods

### 2.1. Standard PCI Method

At present, the PCI method remains the most popular method among engineers related to assessment of the technical condition of airport pavements. The PCI procedure designed and used by the US Army Corps of Engineers has been implemented among institutions such as the Federal Aviation Administration, the US Department of Defense, the American Public Works Association and many others. American standards [27] and [28] were created that systematized the procedures and introduced a catalog of damage. A description of the procedure can also be found in the literature. In the widely known handbook on airport management, road and parking surfaces [8], you can find a detailed description of the PCI index estimation procedure with in-depth commentary.

The PCI indicator is a dimensionless number in the range from 0 to 100 describing the technical condition of the surface, where 100 means the surface is in perfect condition and 0 means the surface is in a completely degraded condition, unfit for any operation. The PCI indicator is determined based on visual inspection of pavement surface damage. During the review, distress type, quantity and severity are taken into account. The scale for assessing the surface based on the PCI index is shown in Figure 5. The same figure also shows the generalized PCI scale for rough surface assessment.

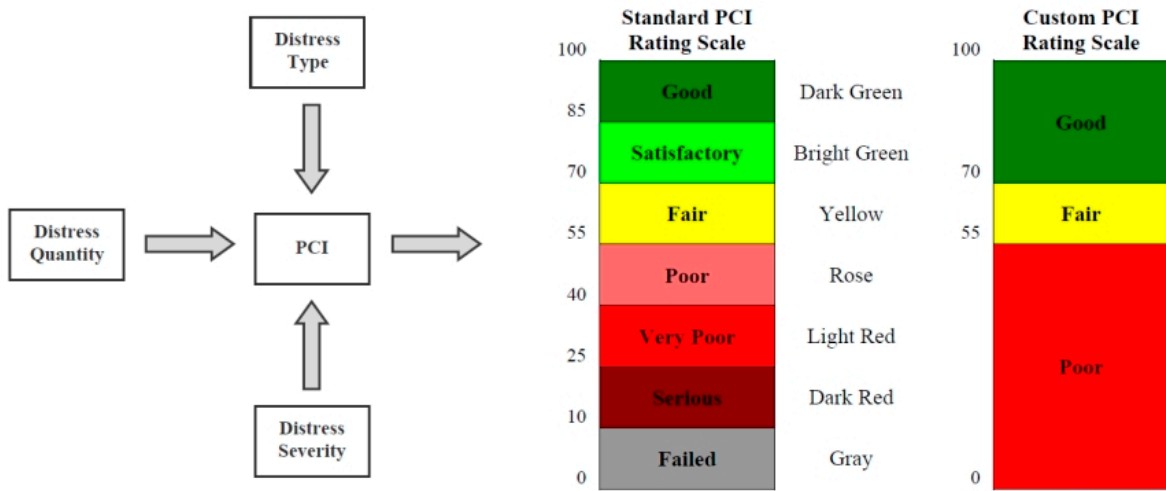

**Figure 5.** Standard and custom PCI Rating Scale. Source: [28].

The surface review is carried out in a strictly defined manner, according to the procedure contained in the standards cited above. The first stage of the work is to divide the airport facility into research samples. The facility is first divided into functional elements of the airport, i.e., runways, taxiways and aprons. Then, each of the elements is divided into individual samples. Division is carried out in

accordance with the instructions. All asphalt surfaces are divided into 5000 ± 1000 sq. ft. samples, which correspond to approximately 460 ± 180 m$^2$. Each sample is considered individually. Due to the fact that testing all the samples is time-consuming, which is reflected in the cost of the review, it is possible to reduce the number of samples. The advantage of this solution is the reduction of costs associated with the review, but at the expense of the quality of results obtained.

The review process itself is based on a list of damages observed on the surface of the pavement according to the legend contained in the standards. The list of damages is made in a table, noting their type, harmfulness, quantity and approximate location on the sample. The number of individual damages, their harmfulness and the density of their occurrence are determined for each analyzed sample. Based on the inventory, the PCI indicator for a particular sample is determined. The final PCI index value for the evaluated element is calculated as the average value of all PCI index values obtained on the analyzed samples. If the samples selected for testing have different surface areas, the weighted average PCI indicator is considered as the final value of the PCI indicator. The surface areas of the individual samples are taken as weights.

### 2.2. APCI Procedure

Assessment of the technical condition of an airport pavement structure should be carried out in a broader context than just based on its surface damage. Air traffic safety is influenced by many factors, including load capacity, anti-skid properties and the evenness of the surface. In addition to the factors mentioned above, the surface tensile bond strength is also important, in particular on airfields where jet-powered aircraft are moving. Due to the durability of the pavement, the load capacity of the structural system, i.e., the entire layer package, is an important parameter. In addition, most repairs to the airport pavement do not restore its original properties 100%. Such repairs should also be considered when assessing the condition of the pavement. A comprehensive approach to assessing the technical condition of the pavement in a systemic way guarantees the management of these pavements in an efficient and sustainable manner. Having the above in mind, a method based on the standard PCI procedure has been developed, extending the existing model with the above-mentioned technical parameters. In this way, a parameter was created in the form of an APCI airport pavement condition indicator (Airfield Pavement Condition Index), describing its comprehensive technical condition. The APCI model takes into account the results of periodic reviews of pavement damages and repairs, load capacity, anti-skid properties, evenness and surface tensile bond strength. The next part presents the assumptions of the developed method, the procedure, and the scale of the APCI indicator, proposed on the basis of many years of research in Polish airport facilities.

Data obtained from various sources may not necessarily provide the same information, in particular those derived from periodic inspections of surface damage and repairs. The same area studied by one expert can be interpreted differently by another. This is influenced by many factors, including the perception of a particular person, degree of fatigue, lighting of the assessed surface, time of day, or even the wetness of the surface. In order for the surface condition assessment to be reliable and largely independent of the human factor, the procedure has been standardized by developing a testing procedure.

Assessment of the condition of airport pavements using the APCI method is based on the results of field tests. A broader picture of the condition of the surface gives the level of the technical condition of the surface, which, in addition to field tests, also includes laboratory tests. Field tests include:

1.  Measuring damage and repairs to the pavement;
2.  Testing the pavement load capacity using the ACN-PCN method (ACN—Aircraft Classification Number; PCN—Pavement Classification Number) in accordance with the requirements of the standard [29];
3.  Testing the skid resistance of the pavement in accordance with the requirements of the standards [30] and [31];
4.  Testing the evenness of the pavement in accordance with requirements of the standard [32];

5.　Surface tensile bond strength (pull-off strength test) in accordance with the requirements of the standard [33].

Meanwhile, laboratory tests include:

1.　Structural tests;
2.　Destructive strength tests;
3.　Climatic tests.

A diagram of the procedure for a comprehensive assessment of the technical condition of the airport functional surface is shown in Figure 6. The basis of the developed methodology is the assessment of individual input parameters based on the results of tests and measurements. When the minimum limits for at least one parameter are not obtained, the assessment is interrupted until the repair is performed. After repairing the non-compliant area, field and laboratory tests should be carried out again. After obtaining all positive test results, the technical condition of the airport pavement is assessed on the basis of field tests and laboratory tests.

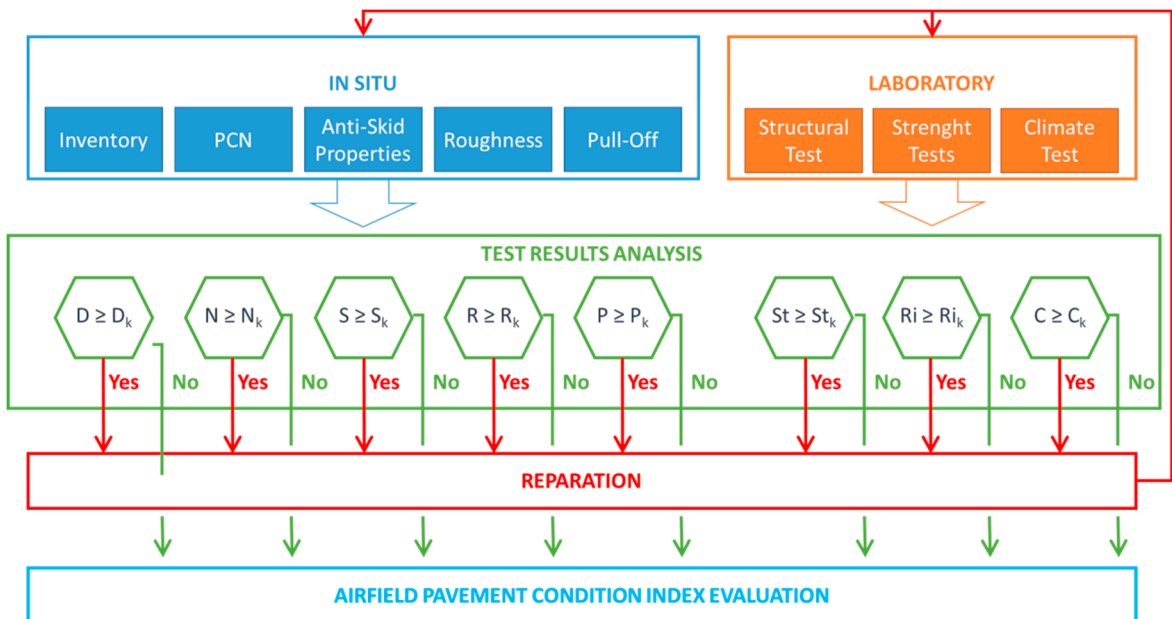

**Figure 6.** Diagram of the procedure for assessing the technical condition of an airport pavement using the APCI indicator.

Measurements and tests carried out in the field are input data for the process of analysis, resulting in output data describing the condition of the airport pavement. The assessment procedure is illustrated in Figure 7.

### 2.2.1. Input Data for APCI Analysis

The data provided for the model of the developed method come from measurements and field tests and are assessment results for degree of degradation, load capacity, anti-skid properties (roughness and texture), evenness and surface tensile bond strength. Each of the above-mentioned tests is conducted in a standardized manner, in accordance with normative documents current as of the date of the test.

Assessment of pavement degradation is performed based on data obtained during the review. Experts measure the damage and repair the surface, taking into account the type, number and location of individual damages. Each of the $10 \times 10$ m virtual panels constituting the assessed functional element of the airport is subject to review. Its location relative to the element is described by two numbers in a row/band arrangement. In addition, systemically, each plate has its unique identifier,

which consists of the airport object symbol in the ICAO notification, the abbreviated name of the airport functional element and the location of the plate. In the case of large elements (e.g., runway), the entire hectometer of the pavement, which corresponds to a fragment of a 100-m-long element, can be assessed. Deteriorations to the panels occur in the form of point, linear and area damage. Point deteriorations include chipping or loosening, blister clusters and wells. Linear deteriorations are primarily cracks, ruts, heaves and fractures. Area deteriorations occur in the form of raveling. The deteriorations legend corresponds to the repair legend, in which the letter markings are identical to the deteriorations. To distinguish deteriorations from repairs, repairs are recorded in black, while deteriorations are marked in red. The amounts of individual damages and repairs are related to the size of the surface on which they were inventoried and expressed in units of pcs/m$^2$, m/m$^2$ or m$^2$/m$^2$ depending on the type of damage. Each type of damage or repair enters the degradation model with a characteristic weight determined by the method of experts based on many years of experience. The degree of degradation of the airport functional element is a dimensionless number from 0 to 100, where undamaged surfaces are characterized by a degradation index of 0, while a completely deteriorated surface has a degradation index of 100.

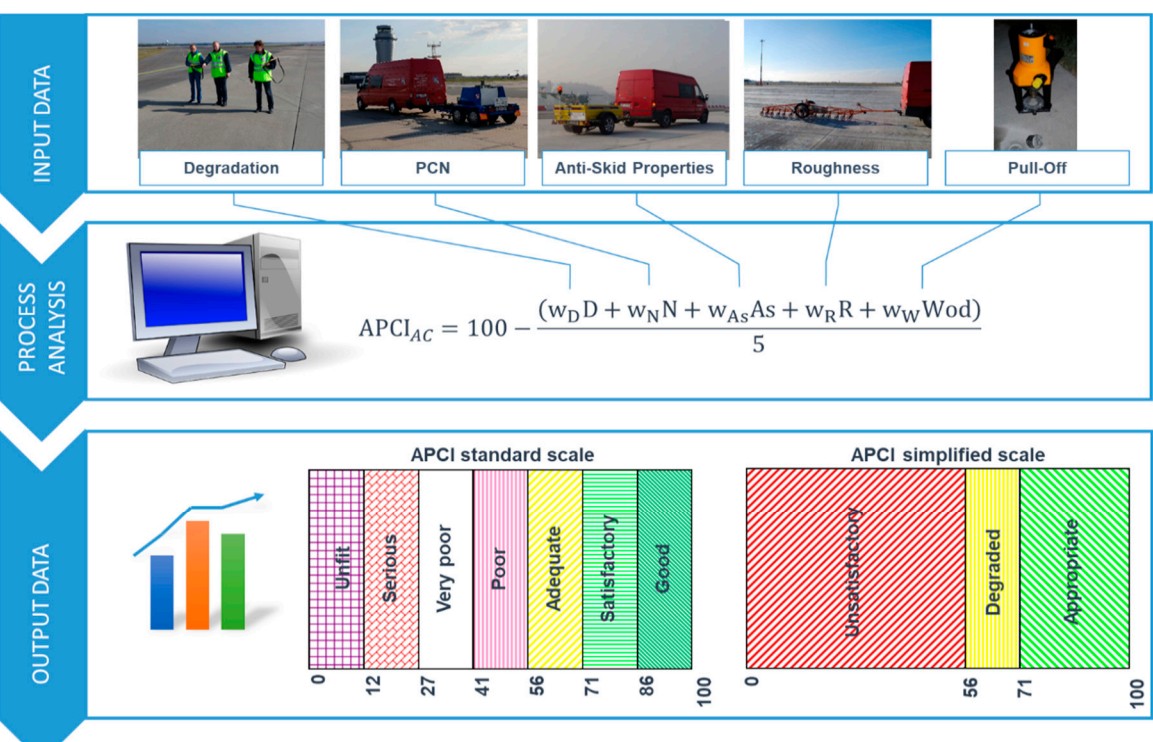

**Figure 7.** Assessment of pavement technical conditions based on the APCI procedure.

Assessment of the load capacity of airport pavement structures is performed using the ACN-PCN method. The test is based on Polish defense standard NO-17-A500: 2016 airfield and road pavement load capacity testing [29]. The values of elastic deflections obtained in the measurement with the Heavy Weight Deflectometer (HWD) device, the pavement construction system, and material parameters of individual layers, including their stiffness (modulus of elasticity) and tensile strength, are taken into account. In addition, the assessment takes into account the subsoil directly under the structure of the evaluated element load parameter. To obtain information about the structure and material parameters of individual layers, full recognition is performed through drilling cores and soil wells and soil subsoil probing. Tensile strength is calculated from the results of tensile strength tests obtained by splitting (using the Brazilian method) cylindrical samples taken from the structural layers of the pavement.

The result of the pavement load capacity assessment is the commonly used PCN indicator and/or the number of permissible air operations for a computational aircraft with the assumed ACN indicator.

The assessment of anti-skid properties is carried out by measuring the coefficient of friction on the airport pavement on the basis of the Polish defense standard NO-17-A501: 2015 Airport Pavements Roughness test and measurement of the pavement texture [30], in accordance with the requirements of the PN-EN 13036-1: 2010 road and airfield surface characteristics test methods: measurement of pavement surface macrotexture depth using a volumetric patch technique [31]. The above standards are in accordance with applicable documents of international aviation organizations, such as the EASA (European Union Aviation Safety Agency), ICAO (International Civil Aviation Organization), and FAA (Federal Aviation Administration). The tests are carried out with a device for continuous measurement of the friction coefficient mentioned in the cited documents. During the test, the friction coefficient is measured at the contact of the braked measuring wheel and the road surface. The test is performed at a speed of 65 or 95 km/h, ensuring a water film thickness in front of the measuring wheel of 1 mm. The results obtained refer to the tables with the requirements contained in the above documents. The requirements relate to the specific measuring devices and specific measuring conditions. The results obtained in testing with one device cannot be compared with the requirements for another device. The test result is a dimensionless number from 0 to 1, where 0 means no friction force, and 1 means friction force equal to the force. The surface texture is measured in two basic ways. The first is the traditional method of a sand patch, in which we obtain the value of the mean texture depth (MTD). It is a point and time-consuming method, requiring appropriate weather conditions and surface cleanliness. The second method involves laser measurement of profile depth. The result of the measurement is the estimated texture depth (ETD) and the mean profile depth (MPD).

The evenness of the airport pavement is assessed on the basis of unevenness measurements, in accordance with the requirements of the Polish defense standard NO-17-A502: 2015 Airfield Pavements Evenness test [32]. The measurement is made with a planograph, which measures and records the clearance values between the theoretical reference line connecting the undersides of the device's road wheels and the surface at the midpoint. The measurement is made at a frequency of 10 cmwith an accuracy of 0.3 mm. The measuring route is divided into 5-m sections, and such sections are subject to assessment. The measure of inequality is defectivity given in percent. This means the percentage of samples exceeding the requirements.

The surface layer tensile bond strength of airport pavements is checked in accordance with the PN-EN 1542: 2000 products and systems for the protection and repair of concrete structures test methods—measurement of bond strength by pull-off—which is also included in the Polish defense standard NO-17-A204: 2015 airport pavements—cement concrete pavements—requirements and test methods [33]. The test is carried out by drilling the surface to a depth of about 15 mm with a lace with a diameter of 50 mm. Then, a disc with a diameter of 50 mm is glued to the surface, and, when the adhesive completely sets, it is broken off using a pull-off apparatus. As a result, a peel strength is obtained which, divided by the surface of the sample, gives the estimated strength.

2.2.2. Process Analysis

Pavement parameters collected during field tests are inputted into the analysis. The developed APCI model for airport asphalt surfaces shows the relationship (1).

$$APCI_{AC} = 100 - (w_D D + w_N N + w_{As} As + w_R R + w_P Wod)/w \qquad (1)$$

where:

| | |
|---|---|
| $w_i$ | characteristic weights for the type of parameter [-], |
| w | sum of weights [-], |
| D | degree of pavement deterioration [%], |
| N | load bearing capacity indicator [%], |
| As | indicator of anti-skid properties [%], |
| R | evenness indicator—longitudinal ($R_L$) and transverse ($R_T$) surface defect [%], |
| Wod | indicator of the surface layer's tensile bond strength [%]. |

The above-mentioned indicators (N, As, Wod) are calculated as percentage consumption. This means that the load bearing capacity (N) is the ratio of the remaining number of operations to the designed number of operations, subtracted from 100%. The anti-skid properties (As) are the ratio of the current state to the required value, subtracted from 100%. The surface layer tensile bond strength (Wod) is the ratio of measured bond strength to the required value, subtracted from 100%.

The weights used in the developed APCI model determination model were selected using the expert method, based on many years of research and experience of engineers involved in the diagnosis of airport pavements. The basis for estimating the weight values was the result of many years of diagnostic tests performed on functional elements of airports in Poland. The tests included both civilian and military facilities on which various types of aircraft operated. Inspections were carried out on all airfield functional elements (AFE), i.e., runways, taxiways and aprons. Periodic inspections covered airport surfaces of various ages and in different conditions, both new and highly degraded, requiring complete restoration.

Comprehensive assessment of the technical condition of the AFE pavement contains the sum of the effects of the above-mentioned parameters, subjected to standardization and loaded with specific weights. The weights of individual parameters are decision variables and depend on the adopted strategy for maintaining the surface. The following three types of strategy are distinguished:

1. Priority to improve the structural condition of the pavement, for which 70% of the total share of parameters was adopted: surface condition (degradation level D) and load capacity (number of permissible air operations N).
2. Priority to improve the condition of air traffic safety, for which 70% of the total share of parameters was adopted: pavement condition (D degradation level) and anti-slip properties (S indicator).
3. Minimizing the cost of maintenance, for which the weights are proportional to the unit costs of maintenance work. This strategy includes parameters that determine the type of maintenance.

Table 1 presents the designated weight values for the adopted AFE pavement maintenance strategies.

**Table 1.** Weights for various systems of AFE pavement parameter combination when calculating the APCI index.

| AFE Pavement Parameters Combination | D [-] | N [-] | As [-] | $R_L$ [-] | $R_T$ [-] | Wod [-] |
|---|---|---|---|---|---|---|
| D; N | 0.40 | 0.60 | | | | |
| D; N; $R_L$; $R_T$ | 0.30 | 0.50 | | 0.10 | 0.10 | |
| D; N; As | 0.30 | 0.55 | 0.15 | | | |
| D; N; $R_L$ | 0.30 | 0.60 | | 0.10 | | |
| D; N; $R_L$; $R_T$; As; Wod | 0.25 | 0.40 | 0.15 | 0.05 | 0.05 | 0.10 |
| D; N; $R_L$; $R_T$; As | 0.30 | 0.40 | 0.15 | 0.05 | 0.10 | |
| D; N; $R_L$; As; Wod | 0.25 | 0.40 | 0.15 | 0.10 | | 0.10 |

### 2.2.3. Output Data

The assessed airport pavements are classified according to the pavement technical assessment criteria, based on the obtained APCI index values. The criteria were developed based on many years of field research.

In order to determine the scale of pavement condition indicators, seven categories of technical condition pavement assessments were introduced for the airport functional element. The entire value range of the selected variable from the minimum to the maximum value is divided into a specified number of sections of equal length. These cases, for which the values of the selected variable belong to one range, form a common category. On a simplified scale, there are three decision levels describing the technical condition of the surface of the airport functional element. Classes determining the surface condition were assigned to each level. The first one is the desired level, which includes new, renovated and exploited surfaces, assuming that within the next five years these surfaces will not require planned renovation works. The warning level, indirect, identifies the condition of the pavement as one in which it is justified to perform detailed tests to carry out treatments to improve the surface condition. The last one is the critical level, determining the immediate performance of technical and operational tests in order to determine the actions aimed at introducing treatments improving the condition of the pavement. Figure 8 presents the relationship of decision levels and AFE pavement technical classes based on the APCI index. Interpretations of pavement classes are detailed in Table 2.

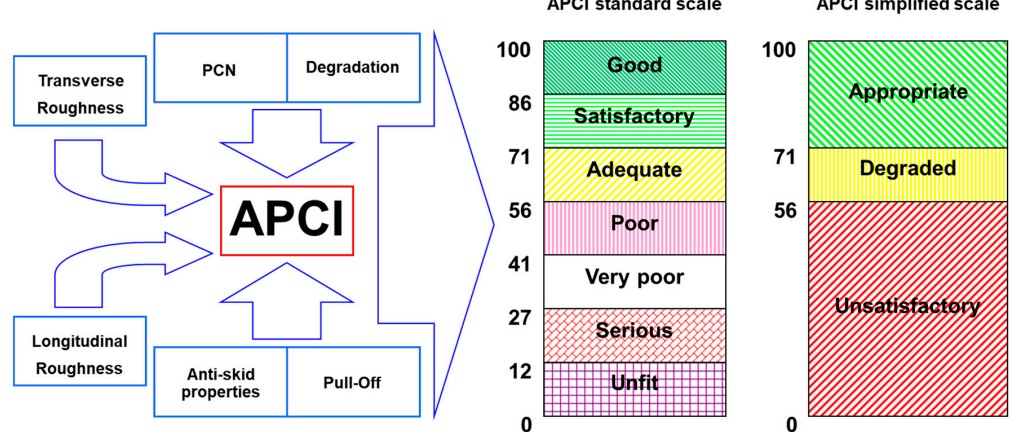

**Figure 8.** Criteria for assessing the technical condition of the surface of the airport functional element by the APCI method.

**Table 2.** Criteria for assessing the technical condition of the AFE pavement.

| Condition | APCI | Definition |
|---|---|---|
| Good | 86 ÷ 100 | The surface is in good technical condition, has little or no damage and only requires routine maintenance. |
| Satisfactory | 71 ÷ 85 | The surface is in satisfactory condition and has damage of low harmfulness, which requires only routine maintenance. |
| Adequate | 56 ÷ 70 | The surface is in good condition and has low and medium damage. Routine and major repairs should be carried out in a short time. |
| Poor | 41 ÷ 55 | The surface is in poor technical condition, and has low-, medium- and high-harmfulness damage, which probably causes operational problems. Maintenance work should include routine repairs and reconstructions in the near future. |
| Very poor | 27 ÷ 40 | The pavement is in very poor technical condition, it has mostly medium- and high-harmfulness damage, which causes significant maintenance and operational problems. Immediate intensive maintenance and repairs are needed. |
| Serious | 12 ÷ 26 | The surface is in a serious technical condition, which usually has damage of high harmfulness, which causes restrictions in its use. Immediate repair is needed. |
| Unfit | 0 ÷ 11 | The pavement is unusable. The deterioration of the technical condition of the pavement has reached a level where safe air operations are no longer possible. Complete reconstruction is necessary. |

## 3. Results

The APCI method was developed as a result of many years of research, expertise and expert panels. The developed model is the result of work financed under the budget of the Ministry of Science and Higher Education. As part of the work, hundreds of tests were carried out at Polish airport facilities, both civilian and military. An extensive database was obtained containing the results of a visual inventory of deteriorations and surface repairs, measurements of elastic deflections, anti-skid properties, evenness and tests of surface tensile bond strength. This article presents a fragment of extensive analysis to illustrate how to use the APCI method to assess the technical condition of asphalt concrete airport pavements. The presented results relate to the operational airport facility in Poland. The airport facility is mainly used by turboprop aircraft. In addition, due to the low tensile bond strength of the surface layer, the movement of jet aircraft is impossible due to the maintenance of flight safety conditions. The presented results refer to the runway (RWY), taxiways (TWY-A, TWY-B, TWY-C, TWY-D, TWY-E and TWY-F) and aprons (APRON A, APRON B, APRON C, APRON D). Due to the confidentiality of the results, the above markings were adopted for the purposes of the article, which do not reflect the actual naming.

An inventory of deteriorations was carried out in accordance with the instructions developed at the Air Force Institute of Technology. The review was based on developed catalogs of airport pavement deteriorations and repairs made of asphalt concrete. Each analyzed functional element of the airport was divided into individual panels. The results were presented graphically on specially prepared underlays. The underlays were in the form of a grid that constitutes the airport's functional element. The grid represented individual panels, and each panel had its individual row/band location. Deterioration was marked with a symbol characterizing the type of damage and a number characterizing the size or harmfulness. The results of the inventory carried out in this way were applied into vector drawings of the airport facility and into the database in tabular form. Based on the results obtained, the degree of pavement degradation was calculated. The procedure has been described, among others, by [21] and [34]. The final result of pavement degradation is described in the following Formulas (2)–(4):

$$D_{AC} = w_{FU} \times W_U + w_{FN} \times W_N \tag{2}$$

$$W_U = \frac{\sum_{i=1}^{11} \frac{(w_U)_i \times (O_U)_i \times (p_U)_i}{F}}{\sum_{i=1}^{11} (w_U)_i} \times 100 \tag{3}$$

$$W_N = \frac{\sum_{i=1}^{11} \frac{(w_N)_i \times (O_N)_i \times (p_N)_i}{F}}{\sum_{i=1}^{11} (w_N)_i} \times 100 \tag{4}$$

where:

$D_{AC}$　　the degree of surface degradation of the asphalt concrete airport functional element [%],
$w_{Fi}$　　statistical weight of the importance of calculated degradation for deteriorations and repairs [-],
$w_i$　　statistical weight of the importance of specific deteriorations and repairs in the AFE degradation assessment [-],
$O_i$　　dimension of deteriorations and repairs of the AFE [depends on type of deterioration: pcs, m, m$^2$],
$p_i$　　conversion of the parameter characterizing deteriorations or repairs to the deteriorated or repaired areas [depends on type of deterioration: m$^2$/pcs; m$^2$/m; m$^2$/m$^2$],
$F$　　total area of the tested AFE pavement [m$^2$],
$U$　　deteriorations to the AFE pavement,
$N$　　repairs of the AFE pavement.

The results of the inventory are presented in the form of degradation of individual elements in Figure 9.

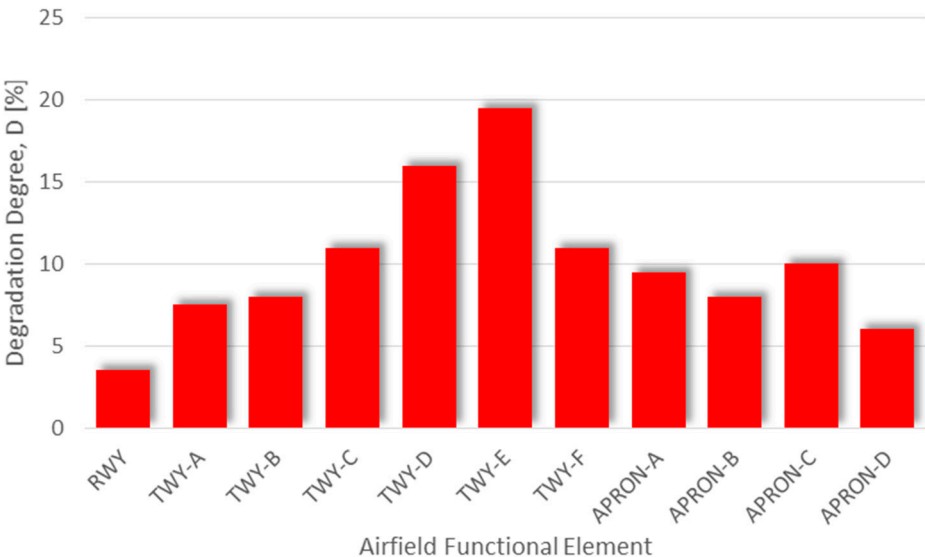

**Figure 9.** Results of pavement degradation.

The elastic deflections were measured each time with an HWD in the airport configuration. At least two measuring routes were determined on each element. The deflection values were saved in a database in tabular form. At the same time, the pavement structure was identified, i.e., the types of layers, their thickness and their physical parameters. Data were obtained from historical research, and, in cases when this was not possible, core drilling was performed. Samples collected in this way were strength-tested. In addition, soil dynamic cone probing was carried out in the drilling hole. In the next step, the PCN value and the allowable number of air operations N for the assumed computational aircraft were determined according to Formula (5):

$$N_{AC} = \left[ \frac{R_{zg}}{\sigma} \times \left( \frac{E}{30000 \text{ MPa}} \right)^{1.3} \right]^{\left( \frac{1}{0.233} \right)} \times 10^4 \tag{5}$$

where:

| | |
|---|---|
| $N_{AC}$ | the number of permissible air operations [-], |
| $R_{zg}$ | ground flexural tensile strength [MPa], |
| $\sigma$ | flexural tensile stress of the ground [MPa], |
| E | soil stiffness modulus [MPa]. |

The results prepared in this way were archived in the database. The load bearing capacity results are shown in Figure 10, in the form of flight operation numbers that can be performed on individual functional elements of the assessed airport facility.

Antiskid properties were measured by measuring the friction coefficient. For this purpose, a friction tester in the form of an ASFT T-10 trailer was used. This is a device for continuous measurement of the friction coefficient accepted by international normative documents. The tests were carried out with water feeding under the measuring wheel in a manner ensuring a water film thickness of 1 mm. All measurements were made at a speed of 65 km/h. In some cases, such as for a runway, in addition, the drive was carried out at a speed of 95 km/h, with only results obtained at a speed of 65 km/h taken into account for the APCI analysis. On each airport functional element, two measurement routes were randomly selected, and the result was the average of both runs. In the case of airport functional elements whose geometrical dimensions did not allow measurements to be taken in safety conditions, the measurement was carried out with the ASFT T2Go hand-held portable friction tester. The device was correlated with ASFT T-10, and the developed mathematical model enabled the conversion of T2Go results into results corresponding to T-10. In this way, it was possible to refer to

the requirements contained in normative documents. The results obtained were saved in a database in tabular form. The result of the measurement was the friction coefficient calculated according to Formula (6):

$$\mu = \frac{T}{N} \tag{6}$$

where:

$\mu$   friction coefficient at the road–wheel contact [-],
$T$   friction force at the wheel–road contact [N],
$N$   pressure force of the measuring wheel on the road surface [N].

The anti-skid property results are shown in Figure 11 in the form of the average coefficient of friction measured with the ASFT T-10 tester.

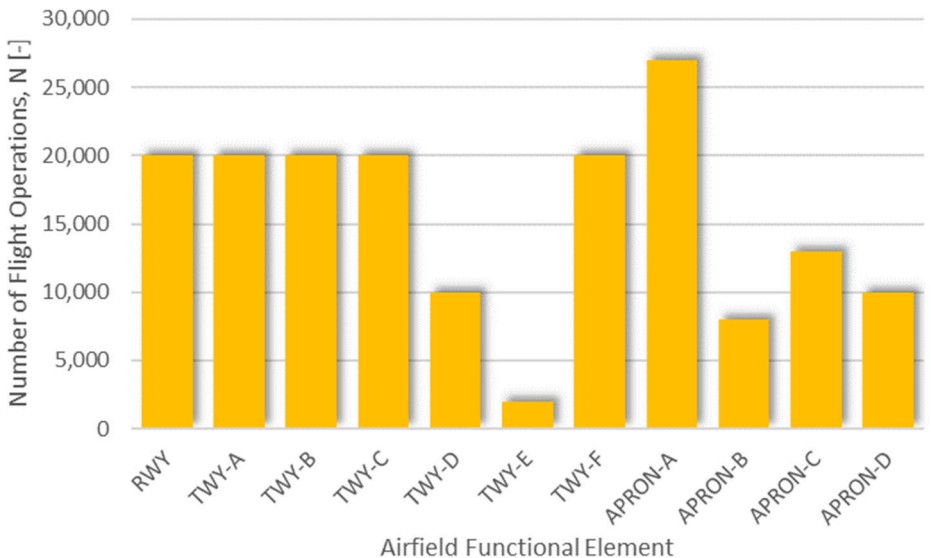

**Figure 10.** Results of the pavement load bearing capacity test.

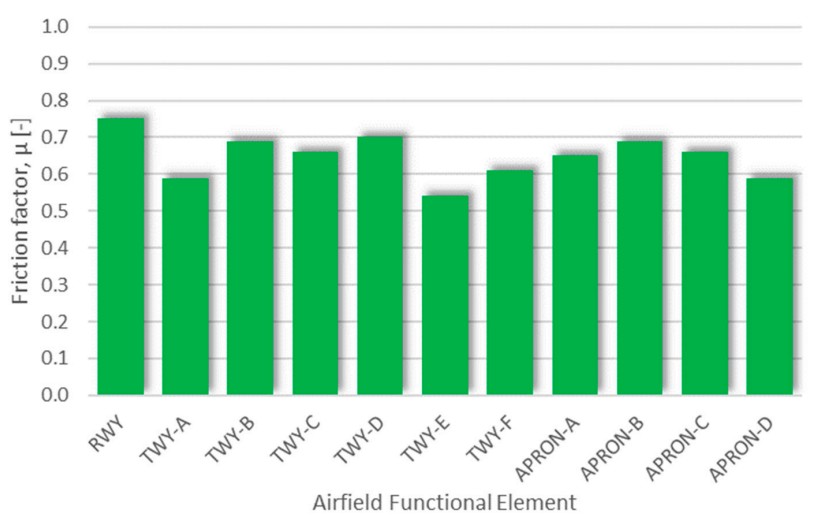

**Figure 11.** Test results for pavement anti-skid properties.

Defectiveness was calculated on the basis of measurement of the unevenness of airport pavements. The measurements were carried out using the P3-z planograph. Each functional element of the airport was divided into 5-m-wide strips, and one measurement run was performed on each. In addition, lateral measurement was carried out on elements wider than 25 m. The distance between the measuring

routes in the transverse direction was 25–50 m. The result of the measurement was the distance between the measuring point and the theoretical reference line, and this was saved as surface unevennesses. Based on the results for unevennesses, the percentage of panels that exceeded the allowable unevenness values was calculated. The defectiveness was calculated according to Formula (7):

$$R = \frac{f}{F} \times 100 \tag{7}$$

where:

| | |
|---|---|
| R | defectiveness [%], |
| f | number of samples exceeding the requirements [-], |
| F | number of all samples [-]. |

In this way, the defect value was obtained for each of the analyzed airport functional elements. The evenness test results are shown in Figure 12 in the form of defects. The graph presents the results of testing both longitudinal and transverse evennesses.

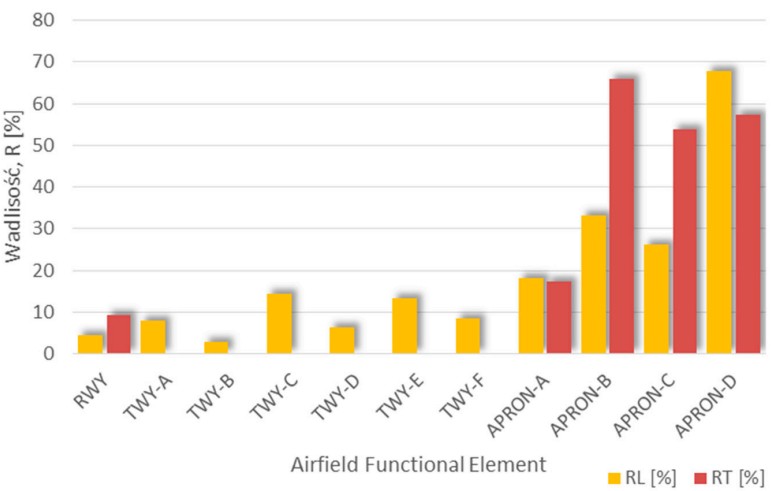

**Figure 12.** Test results of the pavement evenness.

Measurement of the surface tensile bond strength was carried out using the standard method using a pull-off device. The test was carried out at randomly selected points on the airport functional element. To perform the test, a 50-mm-diameter lace was drilled to a depth of about 15 mm. Then, a metal disc with a diameter of 50 mm was glued at the drilling site. After the adhesive had set, the pull-off device was used to pull the disc, thus recording the breaking force. The tensile bond strength value was calculated from Formula (8):

$$Wod = \frac{F}{S} \tag{8}$$

where:

| | |
|---|---|
| Wod | tensile bond strength [MPa], |
| F | maximum breaking force recorded during the measurement [N], |
| S | breaking surface area [mm$^2$]. |

The average of all results for a single functional element of the airfield was the result of the study, which was recorded in the database. Figure 13 presents the results of the study for individual functional elements of the airport.

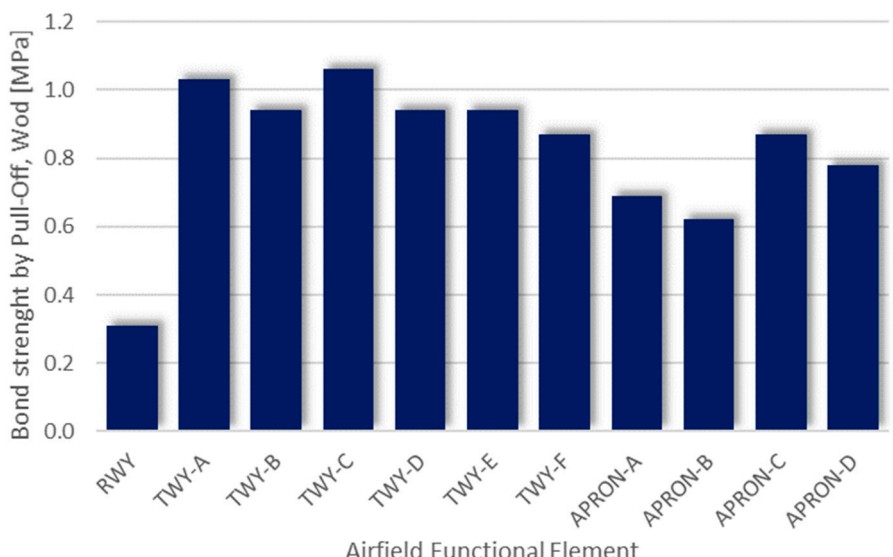

**Figure 13.** Results of testing the surface layer's tensile bond strength.

Table 3 summarizes the results of testing individual parameters with the division into functional elements of the airfield.

Two columns in the table above deserve a comment. The lack of some results in the $R_T$ column is caused by the geometrical dimensions of taxiways, which prevented the measurement of unevenness in the transverse direction. The taxiway width was less than 25 m, as a result of which the measurement was made only in the longitudinal direction. In the second case, in the column containing the results of the surface layer tensile bond strength, the vast majority of results did not meet the required value of 1.0 MPa. In this case, according to the assumptions of the APCI method, corrective actions should be carried out on non-compliant elements, and then reassessment. However, due to the suspended operations of jet aircraft, it was considered that this parameter could be omitted in order to meet the requirements. It should also be noted that the above parameter still occurs in the APCI model, significantly reducing the APCI indicator value. This translates into an assessment of the technical condition of the airport pavement.

**Table 3.** Test results of individual input data parameters for analysis by the APCI method.

| AFE | D [%] | N [-] | As [-] | $R_L$ [%] | $R_T$ [%] | Wod [MPa] |
|---|---|---|---|---|---|---|
| RWY | 3.5 | 20,000 | 0.75 | 4.5 | 9.4 | 0.31 |
| TWY-A | 7.5 | 20,000 | 0.59 | 8.0 | - | 1.03 |
| TWY-B | 8.0 | 20,000 | 0.69 | 3.0 | - | 0.94 |
| TWY-C | 11.0 | 20,000 | 0.66 | 14.3 | - | 1.06 |
| TWY-D | 16.0 | 10,000 | 0.70 | 6.3 | - | 0.94 |
| TWY-E | 19.5 | 2000 | 0.54 | 13.4 | - | 0.94 |
| TWY-F | 11.0 | 20,000 | 0.61 | 8.6 | - | 0.87 |
| APRON-A | 9.5 | 27,000 | 0.65 | 18.3 | 17.3 | 0.69 |
| APRON-B | 8.0 | 8000 | 0.69 | 33.2 | 65.9 | 0.62 |
| APRON-C | 10.0 | 13,000 | 0.66 | 26.2 | 53.7 | 0.87 |
| APRON-D | 6.0 | 10,000 | 0.59 | 67.9 | 57.4 | 0.78 |

Data obtained as a result of measurements and collected in Table 3 enter the APCI model in the form of indicators of individual parameters of input data. In addition, each parameter has a weight assigned to it, adopted by the method of experts based on research results and the knowledge and

experience of specialists in the communications construction industry, in the airport specialization in particular. The calculated indicators are presented in Table 4.

**Table 4.** Indicators of input data parameters for analysis by the APCI method.

| AFE | D [%] | N [%] | As [%] | $R_L$ [%] | $R_T$ [%] | Wod [%] |
|---|---|---|---|---|---|---|
| RWY | 3.5 | 33.3 | 0.0 | 4.5 | 9.4 | 80.6 |
| TWY-A | 7.5 | 33.3 | 15.7 | 8 | | 35.6 |
| TWY-B | 8 | 33.3 | 1.4 | 3 | | 41.3 |
| TWY-C | 11 | 33.3 | 5.7 | 14.3 | | 33.8 |
| TWY-D | 16 | 66.7 | 0.0 | 6.3 | | 41.3 |
| TWY-E | 19.5 | 93.3 | 22.9 | 13.4 | | 41.3 |
| TWY-F | 11 | 33.3 | 12.9 | 8.6 | | 45.6 |
| APRON-A | 9.5 | 10.0 | 7.1 | 18.3 | 17.3 | 56.9 |
| APRON-B | 8 | 73.3 | 1.4 | 33.2 | 65.9 | 61.3 |
| APRON-C | 10 | 56.7 | 5.7 | 26.2 | 53.7 | 45.6 |
| APRON-D | 6 | 66.7 | 15.7 | 67.9 | 57.4 | 51.3 |

The strategy of maintaining the surface, due to the functioning nature of the analyzed airport facility, adopts the priority to improve air traffic safety. For such a solution, 70% of the total share of the parameters pavement condition (D degradation level) and anti-skid properties (As indicator) was assumed. In this context, an assessment was made of the technical condition of airport pavements. Figure 14 presents a bar chart of APCI indicators obtained for individual functional elements of the airport.

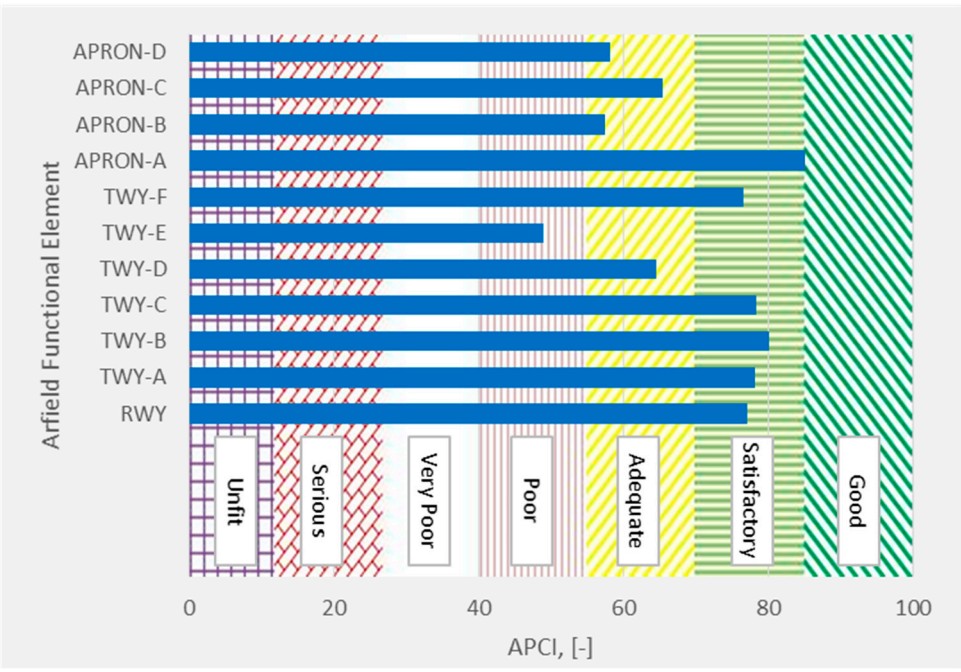

**Figure 14.** APCI values for individual functional elements of the airport.

## 4. Discussion

Proper management of airport asphalt surfaces and related maintenance should be based on reliable information about the technical condition of the surface. Forecasting the technical condition of

the pavement in the future is also important from the point of view of planning funds and resources. Good recognition of changes in the technical condition of the airport pavement is possible only on the basis of a correct assessment of the current technical condition. There are currently many tools that enable this type of assessment. The most popular in the world is the PCI method described earlier and all its types of modifications. The most important of these are described in Section 1. Airport pavement management systems are also based on the PCI method. An example of this type of system is the PAVER system, based on the PCI procedure developed by the US Army Corps of Engineers. As outlined at the beginning, the PCI procedure has its advantages, such as execution time, but it also has its drawbacks. The most important of them is the fact that it only takes into account the results of the inventory of observed surface damage. Other technical parameters of the surface are ignored. To meet the expectations, the authors developed the APCI method, which, in addition to an inventory of asphalt pavement deterioration, also includes the measurement of repairs made and other technical parameters of the pavement. To the other parameters, the authors added surface load bearing capacity, anti-skid properties, and evenness, as well as a very important operational parameter, which is the tensile bond strength of the surface layer. The latter parameter is of great importance for the safety of flight operations, to maintain the cleanliness of airport pavements at the required level.

The research results presented in Section 3 refer to a fragment of an extensive database collected during many years of research. A dozen or so airport facilities were analyzed, at which an inventory of deteriorations and pavement repairs, load capacity, evenness, anti-skid properties of the pavement, and tensile bond strength of the surface layer were tested. The presented results relate to one of the airport facilities where 11 functional elements were tested. The runway, taxiways and parking spaces were tested.

Results obtained were applied to the APCI model through indicators of individual parameters. The indicators have been unified by bringing them to a common unit. Averaging them was possible thanks to this procedure. The weights of individual parameters have been selected using the expert method and introduce input data in a way that characterizes their impact on the technical condition of the pavement.

Based on the test results, the technical condition of the analyzed airport facility was assessed. The lowest result was obtained for the taxiway marked as TWY-E, while the highest was for APRON-A. Four of the assessed functional elements of the airport were distinguished by an APCI index in the range 56–70, which means that the surface is in sufficient technical condition, has damage of low and medium harmfulness, and routine and major repairs should be carried out in a short time. One of the AFEs had an indicator in the range of 41–55, which means that the surface is in poor condition, and has low-, medium- and high-harmfulness damages, which probably cause operational problems. Maintenance work on such surfaces should include routine repairs and reconstructions in the near future, and flight operations should be limited in terms of their safety. Other surfaces, including the runway, have an APCI between 71–86, which means that the surface is in a satisfactory condition, i.e., with low deteriorations that require only routine maintenance.

The development of the APCI model presented in this paper allows for significant automation of the process of assessing the technical condition of asphalt concrete pavements. At the same time, the influence of the human factor on the results of the analysis is reduced. This is important in the airport pavement management cycle in terms of flight safety. In the future, the weight approach presented can easily be replaced by a neural network model. The use of predictive models will allow for a relatively precise estimation of "remaining service life" in the future. The development of a system based on the APCI method will also enable the provision of current information about the surface condition to the relevant services in real time. In addition, collecting data on any parameter can be replaced by an equivalent method at any time, which makes the model extremely versatile.

The next step in the work undertaken is to introduce into the APCI model a parameter related to the direct atmospheric conditions and their impact on the surface. By adding the atmosphere

corrosivity parameter and recognizing its impact on building materials used in the structure, estimation of changes in technical condition will become even more accurate.

## 5. Conclusions

In summary, the currently known method of assessing the technical condition of airport pavements is based only on the pavement surface deterioration inventory. In the world literature, there are many solutions expanding the presented PCI method with additional parameters; however, they are not complete from the point of view of the safety of conducting air operations in the ground maneuvering field. The authors propose an innovative method of assessing the technical condition of airport pavements based on the APCI. The method presented in the article takes into account both surface deteriorations to the pavement as well as repairs carried out. In addition, the model features load bearing capacity, anti-skid properties, evenness of the airport pavement, and, an extremely important parameter, the surface layer's tensile bond strength. The latter parameter is of great importance due to the maintenance of FOD (Foreign Object Debris) principles at the airport facility, in particular where operations are carried out by jet-powered aircraft.

The authors plan to expand the presented method with input data related to the corrosivity of the atmosphere and its impact on airport pavements. As a result, prediction of changes in technical conditions during pavement life will become more accurate and surface management more reliable. This will directly translate into the safety of flight operations and ways of managing financial resources and available resources.

**Author Contributions:** M.W.: conceptualization, methodology, writing—review and editing; P.I.: resources, formal analysis, writing—original draft and editing. All authors have read and agreed to the published version of the manuscript.

**Funding:** Research financed from the budget of the Ministry of Science and Higher Education as part of the statutory activity of the Airfield Division of the Air Force Institute of Technology.

**Conflicts of Interest:** The authors declare no conflict of interest. The funders had no role in the design of the study; in the collection, analyses, or interpretation of data; in the writing of the manuscript; or in the decision to publish the results.

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
