# Peer review of "Evaluation of Asphalt Concrete Airport Pavement Conditions Based on the Airfield Pavement Condition Index (APCI) in Scope of Flight Safety"

_aerospace, doi:10.3390/aerospace7060078_

Round 1

Reviewer 1 Report

It is my belief that the authors rearrange the article in a fair direction, being more informative and sharper within the description of the methodology. Some aspects continue to be present (better but still some dense and wordy descriptions that don’t allow a fluid reading). Some also better explanation was given when comparing to other methodologies, avoiding however, in some extent, the cost of implementation issues. The technical description is now more understandable and with a proper format.

I also believe that the contribution how it is now is worthy to readers and describing a process (now having a better support) it is also worthy to be evaluate by Airport managers.

Author Response

Thank you for revision. Some small changes were made.

Reviewer 2 Report

The authors are commended for a well written paper and presenting an overall asphalt pavement condition index incorporating additional technical parameters of importance to airport pavements.

The APCI requires more and resource intensive inputs that PCI.  How often (years) the pavements needs to surveyed/assessed?  For example, FAA Advisory Circular 150/5380-7B notes "If a pavement condition index (PCI) survey is performed, as set forth in ASTM D5340, Standard Test Method for Airport Pavement Condition Index Surveys, the frequency of the detailed inspections by PCI surveys may be extended to three years."

While an overall condition indices, such as PCI or APCI, is useful for rating and communication, are they still the the best for forecasting or, for example calculating RSL?  For illustration, assume an overall condition index based on two defects (x1 and x2) in % and with equal weights. Overall condition index of 50 will be obtained for x1 = 50% and x2 = 50% or x1 = 25% and x2 = 75% but the RSL for latter will be lower than the former since x1 will reach terminal for latter sooner.

Please see attached PDF for specific and editorial comments.

Author Response

Thank you for revision.

It is good practise to provide pavement evaluation at least one time during 5 years of exploatation. Naturaly the more frequent evaluation will be more accurate and forecasting of remaining service life will be more precise.

Your specific and editorial comments were implemented.

This manuscript is a resubmission of an earlier submission. The following is a list of the peer review reports and author responses from that submission.

Round 1

Reviewer 1 Report

Let me start to say that the article has a straightforward objective and to achieve it you follow up a fair framework. The article has dense and wordy descriptions that don’t allow a fluid reading and has some basic format flaws as units for variables in each equation described. The abstract should be improved because is not meaningful when compared with the main issues treated in the article. Also is not clear the origin of the process described neither the origin and variability of the data used to validate it, how validation was performed, and no information about the data to demonstrate APCI application. How can the practitioner rely on the description without knowing details about the approach development? In this sense, it is my belief that you need to show and discuss these aspects unless the article is more a commercial (or institutional?) presentation of a tool to deal with airport pavement conditions. Another thing that you should do is addressing the time and resources consuming that the tool implies when compared with ASTM PCI procedure, for instance (there are other processes). You need to explain the overall concept unless it can be seen as not feasible for some “on time” maintenance decisions. In my view, the article is not ready to be published in the context of a research and application forum.

Reviewer 2 Report

  1. Please rewrite the abstract and introduction which seem too long.
  2. Is it necessary to spend about 2 pages to introduce a well-known PCI?
  3. Please include a full name for “CAN-PCN” in line 287 when it first appears.
  4. What’s “climatic tests” in line 297?
  5. Please double check the format of this manuscript. For example, in line 333, “expressed in units pcs / m2”. In Table 2, APCI values “86 ÷ 100”?